# Chiral Perturbation Theory vs. Linear Sigma Model in a Chiral Imbalance Medium

**Alexander Andrianov** [1,2,*,†], **Vladimir Andrianov** [1,†] **and Domenec Espriu** [2,†]

[1] V. A. Fock Department of Theoretical Physics, Saint-Petersburg State University, 199034 St. Petersburg, Russia; v.andriano@rambler.ru

[2] Departament de Física Quàntica i Astrofísica and Institut de Ciències del Cosmos (ICCUB), Universitat de Barcelona, Martí i Franquès 1, 08028 Barcelona, Spain; espriu@icc.ub.edu

\* Correspondence: andrianov@spbu.ru; Tel.: +7-921-44-35355

† These authors contributed equally to this work.

**Abstract:** We compare the chiral perturbation theory (ChPT) and the linear sigma model (LSM) as realizations of low energy quantum chromodynamics (QCD) for light mesons in a chirally-imbalanced medium. The relations between the low-energy constants of the chiral Lagrangian and the corresponding constants of the linear sigma model are established as well as the expressions for the decay constant of $\pi$-meson in the medium and for the mass of the $a_0$. In the large $N_c$ count taken from QCD the correspondence of ChPT and LSM is remarkably good and provides a solid ground for the search of chiral imbalance manifestations in pion physics. A possible experimental detection of chiral imbalance (and therefore a phase with local parity breaking) is outlined in the charged pion decays inside the fireball.

**Keywords:** chiral imbalance; chiral perturbation theory; linear sigma model; charged pion decay in chiral medium; local parity breaking

## 1. Introduction

The possible generation of a phase with local parity breaking (LPB) in nuclear matter at extreme conditions such as those reached in heavy ion collisions (HIC) at the Relativistic Heavy Ion Collider (RHIC) and the Large Hadron Collider (LHC) [1] has been examined recently [2–8]. It has been suggested in [2–5] that at increasing temperatures an isosinglet pseudoscalar background could arise due to large-scale topological charge fluctuations (studied recently in lattice quantum chromodynamics (QCD) simulations [9–11]).

These considerations led eventually to the observation of the so-called chiral magnetic effect (CME) [2–5] in the STAR and PHENIX experiments at RHIC [12,13]. The effect should be most visible for non-central HIC where large angular momenta induce large magnetic fields contributing to the chiral charge separation. However, the CME may be only a partial explanation of the STAR and PHENIX experiments and other backgrounds play a comparable role (see the reviews [14–17]). In a recent report [18] the measurements of the chiral magnetic effect in Pb–Pb collisions with A Large Ion Collider Experiment (ALICE) were estimated and perspectives to improve their precision in future LHC runs were outlined.

For central collisions it was proposed in [6,7] that the presence of a phase where parity was spontaneously broken could be a rather generic feature of QCD. Local parity breaking can be induced by difference between the densities of the right- and left-handed chiral fermion fields (chiral Imbalance) in metastable domains with non-zero topological charges. Thus our analysis concerns solely the events in the central heavy ion collisions where the magnetic fields are negligible. It is seen in the

experiments [12–17], and was also found in lattice QCD (see [9–11]). The validity of CME and its percentage in observations is well analyzed in [18]. Thereby the elimination of electromagnetic effects is justified and allows to measure solely the chiral chemical potential without contamination by magnetic fields and related backgrounds.

In the hadron phase we shall assume that as a consequence of topological charge fluctuations, the environment in the central HIC generates a pseudoscalar background growing approximately linearly in time. This background is associated with a constant axial vector whose zero component is identified with a chiral chemical potential. In such an environment one could search for a possible manifestation of LPB in dilepton probes. In particular, in [19,20] it was shown that a good part of the excess of dileptons produced in central heavy-ion collisions [21] might be a consequence of LPB due to the generation of a pseudoscalar isosinglet condensate whose precise magnitude and time variation depends on the dynamics of the HIC.

The complete description of a medium with chiral imbalance should also take into account thermal fluctuations of the medium. In this paper the description in a zero temperature limit is considered and to understand the changes for non-zero temperatures we rely on the results of lattice computations of quark matter with chiral imbalance and a temperature of order 150 MeV undertaken [22,23]. Thus our calculations keep the tendency of increasing chiral condensate and decreasing pion masses when the temperature grows.

This paper is mostly concerned with the possibility of identifying LPB in the hadron phase of QCD in HIC. Such a medium would be simulated by a chiral chemical potential $\mu_5$. Adding to the QCD Lagrangian the term $\Delta\mathcal{L}_q = \mu_5 q^\dagger \gamma_5 q \equiv \mu_5 \rho_5$, we allow for non-trivial topological fluctuations [19,20] in the nuclear (quark) fireball, which are ultimately related to fluctuations of gluon fields. The transition of the quark–gluon medium characteristics to a hadron matter reckons on the quark–hadron continuity [24] after hadronization of quark–gluon plasma. The behavior of various spectral characteristics for light scalar and pseudoscalar $(\sigma, \pi^a, a_0^a)$-mesons by means of a QCD-motivated $\sigma$-model Lagrangian was recently derived for $SU_L(2) \times SU_R(2)$ flavor symmetry including an isosinglet chiral chemical potential [25,26]. The structural constants of the $\sigma$-model Lagrangian were taken as input parameters suitable to describe the light meson properties in vacuum and then they are extrapolated to a chiral medium. In this way ad hoc there is no reliable predictability in the determination of the hadron system response on chiral imbalance, and reaching quantitative predictions requires a phenomenologically justified hadron dynamics. To increase predictability, we extend the vacuum chiral Lagrangians [27–29] with phenomenological low-energy structural constants taking into account the chiral medium in the fireball with a chiral imbalance. It is shown that $\sigma$-model parametrization of [25,26] fits well the pion phenomenology at low energies as derived from ChPT.

Next it is described how pions modify their dynamics in decays in a chiral medium, in particular, charged pions stop decaying into muons and neutrinos for a large enough chiral chemical potential. A possible experimental detection of chiral imbalance (and therefore a phase with local parity breaking) is outlined in the charged pion decays inside the fireball.

## 2. Chiral Lagrangian with Chiral Chemical Potential

The chiral Lagrangian for pions describing their mass spectra and decays in the fireball with a chiral imbalance can be implemented with the help of softly broken chiral symmetry in QCD transmitted to hadron media, a properly constructed covariant derivative:

$$D_\nu \Longrightarrow \bar{D}_\nu - i\{\mathbf{I}_q \mu_5 \delta_{0\nu}, \star\} = \mathbf{I}_q \partial_\nu - 2i\mathbf{I}_q \mu_5 \delta_{0\nu}, \tag{1}$$

where we skipped the electromagnetic field. The axial chemical potential is introduced as a constant time component of an isosinglet axial-vector field.

In the framework of large number of colors $N_c$ [29] the SU(3) chiral Lagrangian in the strong interaction sector contains the following dim=2 operators [29],

$$\mathcal{L}_2 = \frac{F_0^2}{4} < -j_\mu j^\mu + \chi^\dagger U + \chi U^\dagger >, \tag{2}$$

where $< ... >$ denotes the trace in flavor space, $j_\mu \equiv U^\dagger \partial_\mu U$, the chiral field $U = \exp(i\hat{\pi}/F_0)$, the bare pion decay constant $F_0 \simeq 92$ MeV, $\chi(x) = 2B_0 s(x)$ and $M_\pi^2 = 2B_0 \hat{m}_{u,d}$, the tree-level neutral pion mass. The constant $B_0$ is related to the chiral quark condensate $< \bar{q}q >$ as $F_0^2 B_0 = - < \bar{q}q >$. Taking now the covariant derivative in (1) it yields

$$\mathcal{L}_2(\mu_5) = \mathcal{L}_2(\mu_5 = 0) + \mu_5^2 N_f F_0^2. \tag{3}$$

Herein we have used the identity for $U \in SU(n)$, $< j_\mu >= 0$. In the large $N_c$ approach the dim=4 operators [29] in the chiral Lagrangian are given by

$$\mathcal{L}_4 = \bar{L}_3 < j_\mu j^\mu j_\nu j^\nu > + L_0 < j_\mu j_\nu j^\mu j^\nu > -L_5 < j_\mu j^\mu (\chi^\dagger U + \chi U^\dagger) >, \tag{4}$$

where $L_0, \bar{L}_3, L_5$ are bare low energy constants. For SU(3) and SU(2) $< j_\mu >= 0$ and there is the identity

$$< j_\mu j_\nu j^\mu j^\nu >= -2 < j_\mu j^\mu j_\nu j^\nu > + \frac{1}{2} < j_\mu j^\mu >< j_\nu j^\nu > + < j_\mu j_\nu >< j^\mu j^\nu >, \tag{5}$$

whereas for SU(2) there is one more identity

$$2 < j_\mu j^\mu j_\nu j^\nu >=< j_\mu j^\mu >< j_\nu j^\nu > . \tag{6}$$

Applying these identities one finds the four-derivative Gasser–Leutwyler (GL) operators for the SU(3) chiral Lagrangian

$$\begin{aligned} \mathcal{L}_4 \quad = \quad & L_1 < j_\mu j^\mu >< j_\nu j^\nu > + L_2 < j_\mu j_\nu >< j^\mu j^\nu > + L_3 < j_\mu j^\mu j_\nu j^\nu > \\ & - L_5 < j_\mu j^\mu (\chi^\dagger U + \chi U^\dagger) > \end{aligned} \tag{7}$$

with

$$L_1 = \frac{1}{2} L_0; \; L_2 = L_0; \; L_3 = \bar{L}_3 - 2L_0. \tag{8}$$

For SU(2) one has a further reduction of the dim=4 Lagrangian,

$$\mathcal{L}_4 = \frac{1}{4} l_1 < j_\mu j^\mu >< j_\nu j^\nu > + \frac{1}{4} l_2 < j_\mu j_\nu >< j^\mu j^\nu > - \frac{1}{4} l_4 < j_\mu j^\mu (\chi^\dagger U + \chi U^\dagger) > \tag{9}$$

with normalization so that

$$l_1 = 4L_1 + 2L_3, \; l_2 = 4L_2, \; (l_1 + l_2) = 2L_3 + 6L_2; \quad l_4 = 4L_5. \tag{10}$$

We stress that this chain of transformations is valid only if $< j_\mu >= 0$.

The response of the chiral Lagrangian on chiral imbalance is derived with the help of the covariant derivative (1) applied to the Lagrangian (4),

$$\Delta \mathcal{L}_4(\mu_5) = -\mu_5^2 \{8(l_1 + l_2) < j^0 j^0 > -4(l_1 + l_2) < j_k j_k > -l_4 < \chi^\dagger U + \chi U^\dagger >\}. \tag{11}$$

We notice that this result is drastically different from what one could obtain from the final Lagrangian (9). This is because the identities (5) and (6) are violated if $< j_\mu >\neq 0$. The above modifications change differently the coefficients in the dispersion law in energy $p^0$ and three-momentum

$|\vec{p}|$ for the mass shell as well as modify the mass term for pions (all together it gives the inverse propagator of pions),

$$\mathcal{D}^{-1}(\mu_5) = (F_0^2 + 32\mu_5^2(l_1 + l_2))p_0^2 - (F_0^2 + 16\mu_5^2(l_1 + l_2))|\vec{p}|^2 - (F_0^2 + 4l_4\mu_5^2)m_\pi^2 \to 0. \quad (12)$$

In the leading order of large $N_c$ expansion (neglecting the renormalization group (RG) logarithm as a contribution next-to-leading in the large $N_c$ expansion ) the empirical values of the SU(2) Gasser-Leutwyler (GL) constants [27,28] are

$$l_1^r = (-0.4 \pm 0.6) \times 10^{-3}; \ l_2^r = (8.6 \pm 0.2) \times 10^{-3};$$
$$l_1^r + l_2^r = (8.2 \pm 0.8) \times 10^{-3}; \ l_4^r = (2.64 \pm 0.01) \times 10^{-2}. \quad (13)$$

They can be obtained also if they are normalized at the renormalization group scale $\mu \simeq M_\pi \simeq 140$ MeV, $\log\left(m_\pi/\mu\right) \simeq 0$.

Thus in the pion rest frame

$$F_\pi^2(\mu_5^2) \simeq F_0^2 + 32\mu_5^2(l_1 + l_2); \quad m_\pi^2(\mu_5^2) \simeq \left(1 - 4\frac{\mu_5^2}{F_0^2}(8(l_1 + l_2) - l_4)\right)m_\pi^2(0), \quad (14)$$

i.e., the pion decay constant is growing and its mass is decreasing in the chiral media.

### 3. Linear Sigma Model for Light Pions and Scalar Mesons in the Presence of Chiral Imbalance: Comparison to ChPT

Let us compare these constants with those ones estimated from the linear sigma model (LSM) built in [30–32]. The sigma model was build with realization of SU(2) chiral symmetry to describe pions and isosinglet and isotriplet scalar mesons. Its Lagrangian reads

$$L = N_c \left\{ \frac{1}{4} < (D_\mu H (D^\mu H)^\dagger > + \frac{B_0}{2} < m(H + H^\dagger > + \frac{M^2}{2} < HH^\dagger > \right.$$
$$\left. - \frac{\lambda_1}{2} < (HH^\dagger)^2 > - \frac{\lambda_2}{4} < (HH^\dagger) >^2 + \frac{c}{2} (\det H + \det H^\dagger) \right\}, \quad (15)$$

where $H = \xi \Sigma \xi$ is an operator for meson fields, $N_c$ is a number of colours, $m$ is an average mass of current $u, d$ quarks, $M$ is a "tachyonic" mass generating the spontaneous breaking of chiral symmetry, $B_0, c, \lambda_1, \lambda_2$ are real constants.

The matrix $\Sigma$ includes the singlet scalar meson $\sigma$, its vacuum average $v$ and the isotriplet of scalar mesons $a_0^0, a_0^-, a_0^+$, the details see in [30–32]. The covariant derivative of $H$ including the chiral chemical potential $\mu_5$ is defined in (1). The operator realizes a nonlinear representation (see (2)) of the chiral group $SU(2)_L \times SU(2)_R$, namely, $\xi^2 = U$.

The diagonal masses for scalar and pseudoscalar mesons read

$$m_\sigma^2 = -2\left(M^2 - 6(\lambda_1 + \lambda_2)F_\pi^2 + c + 2\mu_5^2\right)$$
$$m_a^2 = -2\left(M^2 - 2(3\lambda_1 + \lambda_2)F_\pi^2 - c + 2\mu_5^2\right)$$
$$m_\pi^2(\mu_5) = \frac{2bm}{F_\pi} \simeq m_\pi^2(0)\left(1 - \frac{\mu_5^2}{2(\lambda_1 + \lambda_2)F_0^2}\right) \quad (16)$$
$$F_\pi^2(\mu_5) = \frac{M^2 + 2\mu_5^2 + c}{2(\lambda_1 + \lambda_2)} = F_0^2 + \frac{\mu_5^2}{\lambda_1 + \lambda_2}.$$

From spectral characteristics of scalar mesons in vacuum one fixes the Lagrangian parameters, $\lambda_1 = 16.4850$, $\lambda_2 = -13.1313$, $c = -4.46874 \times 10^4$ MeV$^2$, $B_0 = 1.61594 \times 10^5$ MeV$^2$ [30,31].

The change of the pion-coupling constant $F_0$ is determined by potential parameters as compared to the ChPT definition,

$$\frac{\Delta F_\pi^2}{\mu_5^2} = \frac{1}{\lambda_1 + \lambda_2} \approx 0.3 \quad vs \quad 32(l_1 + l_2) \approx 0.26. \tag{17}$$

It is a quite satisfactory correspondence.

Analogously, in the rest frame using the pion mass correction, $m_\pi^2(\mu_5) F_\pi^2(\mu_5) \simeq 2 m_q B_0 F_\pi(\mu_5)$ it is easy to find the estimation for

$$l_4 \approx 2.64 \times 10^{-2} \quad vs \quad \frac{1}{8(\lambda_1 + \lambda_2)} \approx 3.8 \times 10^{-2}, \tag{18}$$

wherefrom one can also guess the relation $4(l_1 + l_2) \sim l_4$ following from the LSM.

For moving mesons with $|\vec{p}| \neq 0$ and the CP breaking mixing of scalar and pseudoscalar mesons the effective masses $m_{eff\mp}^2$ take the form,

$$\begin{aligned}
m_{eff-}^2 &= \frac{1}{2}\left(16\,\mu_5^2 + m_a^2 + m_\pi^2 - \sqrt{(16\mu_5^2 + m_a^2 + m_\pi^2)^2 - 4\left(m_a^2\,m_\pi^2 - 16\mu_5^2\,|\vec{p}|^2\right)}\,\right), \\
m_{eff+}^2 &= \frac{1}{2}\left(16\,\mu_5^2 + m_a^2 + m_\pi^2 + \sqrt{(16\mu_5^2 + m_a^2 + m_\pi^2)^2 - 4\left(m_a^2\,m_\pi^2 - 16\mu_5^2\,|\vec{p}|^2\right)}\,\right).
\end{aligned} \tag{19}$$

For small $\mu_5^2, m_\pi^2 \ll m_a^2 \simeq 1 GeV^2$ one can approximate the dependence on the wave vector $\vec{p}$

$$m_{eff-}^2 \simeq m_\pi^2 - 16\mu_5^2 \frac{|\vec{p}|^2}{m_a^2}. \tag{20}$$

Comparing with (12) one establishes the relationship of isotriplet scalar mass and GL constants

$$m_a = \frac{F_0}{\sqrt{l_1 + l_2}} \simeq 1 GeV, \tag{21}$$

which reproduces the Particle Data Group (PD) value within the experimental error bars [33].

## 4. Possible Experimental Detection of Chiral Imbalance in the Charged Pion Decays

The predicted distortion of the mass shell condition can be detected in decays of charged pions when the effective pion mass approaches muon mass. Let us find the threshold value for the $\pi^+ \to \mu^+\nu$ decay. If a charged pion was generated in chiral medium its mass is lower than in the vacuum and the condition for its decay follows from (12),

$$\left(1 - 16(l_1 + l_2)\frac{\mu_5^2}{F_0^2}\right)\left(|\vec{p}|^2 + m_{0,\pi}^2\right) \geq |\vec{p}|^2 + m_\mu^2, \quad \frac{m_a^2}{16\mu_5^2} \geq \frac{|\vec{p}|^2 + m_{0,\pi}^2}{m_{0,\pi}^2 - m_\mu^2}, \tag{22}$$

where we have used the relations $4(l_1 + l_2) \simeq l_4$ and (21). The decay channel is closed for $|\vec{p}|^2 \simeq 0$ if $\mu_5 \simeq 160$ MeV. It must be detected as a substantial decrease of muon flow originated from pion decays in the fireball. When considering the decay process of a charged pion into a muon + neutrino at values of the chiral chemical potential lower than $\mu_5 \simeq 160$ MeV then still the muon yield from the fireball obviously decreases at sufficiently large momenta. It gives one a chance to measure the magnitude of chiral chemical potential for sufficiently high statistics.

## 5. Results

- For light mesons in the chiral imbalance medium we compared the chiral perturbation theory (ChPT) and the linear sigma model (LSM) as realizations of low energy QCD. The relations between the low-energy constants of the chiral Lagrangian and the corresponding constants of the linear sigma model are established and expressions for the decay constant of the pion in the medium and the mass of the $a_0$ meson are found.

- The low energy QCD correspondence of ChPT and LSM in the large $N_c$ limit is satisfactory and provide a solid ground for the search of chiral imbalance manifestation in pion physics at HIC.

- The resulting dispersion law for pions in the medium allows us reveal the threshold of decay of a charged pion into a muon and neutrino which can be suppressed by increasing chiral chemical potential.

- As it is shown in [30,31], at higher energies exotic decays of isoscalar mesons into three pions arise due to mixing of $\pi$ and $a_0$ meson states in the presence of chiral imbalance. It was shown [19,20,25,26,30,31] that for a wider class of direct parity breaking at higher energies, in the framework of linear sigma model with isotriplet scalar ($a_0$) and pseudoscalar (pions) mesons, their mixing arises with the generation of $\pi\pi$ and $\pi\pi\pi$ decays of a heavier scalar state. Also, the independent check of our estimates could be done by lattice computation (cf. [22,23]).

- A manifestation for LPB can also happen in the presence of chiral imbalance in the sector of $\rho$ and $\omega$ vector mesons [6–8] and in this case the Chern–Simons interaction plays a major role. It turns out [6,7] that the spectrum of massive vector mesons splits into three components with different polarizations $\pm$, *long* having different effective masses $m_{V,+} < m_{V,long} < m_{V,-}$.

- Thus a possible experimental detection of chiral imbalance in medium (and therefore a phase with LPB) in the charged pion decays and vector meson polarizations inside of the fireball can be realized.

- We would like to mention the recent proposal to measure the photon polarization asymmetry in $\pi\gamma$ scattering [30,31,34,35] as a way to detect LPB due to chiral imbalance. This happens in the ChPT including electromagnetic fields due to the Wess–Zumino–Witten operators.

- One may be concerned about the appearance of changes in the properties of muons and neutrinos in the medium, but in our opinion, this does not change the main estimates in Equation (22), as a possible influence of chiral chemical potential on lepton properties would be controlled by extra power of the Fermi coupling constant, i.e., by the next order in weak interactions with little hope to register it.

- We emphasize the similarity of our model results to lattice computation in [22,23]: to the same tendency of increasing chiral condensate and decreasing of pion mass when $\mu_5$ grows for fixed temperatures about 150 MeV. It gives us the confidence (see [25,26,30,31]) that our spectral predictions are robust in a range of temperatures. We understand that for a more realistic quantitative description of the phenomena under discussion, thermal effects, smearing of data and detector acceptance must be taken into account, which will be done in subsequent works with an extended team including the experimentalists.

- Last decade, different controversial conclusions on thermodynamics of quark matter with a chiral imbalance appeared based on different models of the Nambu-Jona-Lasinio (NJL) type. Among them, an opposite decreasing behavior of quark condensate in [36,37] was found due to an erroneous use of UV regularization in a NJL type model which mimicked chiral symmetry breaking, and chiral imbalance with chiral chemical potential being included into an UV cutoff. This kind of mistake in applications of NJL models has been known since the 1980s. The correct regularization based on vacuum definitions of cutoffs is elucidated in [38]. In the treatment in [22,23] based on lattice computations as well as in meson Lagrangians [32] where the UV finite chirally-symmetric computations are used, the problem is thoroughly resolved.

**Author Contributions:** Data curation, D.E.; Investigation, A.A., V.A. and D.E.; Methodology, V.A.; Writing—original draft, A.A. All authors have read and agreed to the published version of the manuscript.

**Funding:** This research was funded by the Grant FPA2016-76005-C2-1-P and Grant 2017SGR0929(Generalitat de Catalunya), by Grant RFBR 18-02-00264 and by SPbSU travel grants Id: 43179834, 43178702, 43188447, 41327333, 36273714, 41418623.

**Acknowledgments:** We express our gratitude to Angel Gómez Nicola for stimulating discussions of how to implement chiral imbalance in ChPT.

**Conflicts of Interest:** The authors declare no conflict of interest.

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
