# Peer review of "Chiral Perturbation Theory vs. Linear Sigma Model in a Chiral Imbalance Medium"

_2571-712X, doi:10.3390/particles3010002_

Round 1

Reviewer 1 Report

This paper studies pions (and other light mesons) within two well-established realizations of the low-energy QCD, namely the chiral perturbation theory and the linear sigma model.

The authors take into account the possibility of chiral imbalance in such a medium. The effects of such chiral imbalance in the heavy-ion-physics of pions are explored. 

The paper is a short and well-written contribution to the subject and I recommend its publication in the Particles.

Author Response

 Point I:

This paper studies pions (and other light mesons) within two well-established realizations of the low-energy QCD, namely the chiral perturbation theory and the linear sigma model.

The authors take into account the possibility of chiral imbalance in such a medium. The effects of such chiral imbalance in the heavy-ion-physics of pions are explored.

The paper is a short and well-written contribution to the subject and I recommend its publication in the Particles.

Response I:    We agree with  the Reviewer 1 Comments

Reviewer 2 Report

This paper aims to find a way to identify a chiral imbalance in heavy ion experiments. They consider XPT and LSM in a chirally imbalanced medium concluding they have found good agreement between the two theories in predicting a change in the effective pion mass that can then be detected in the decays of charged pions.

I have several comments:

The authors claim in the second paragraph of the introduction that the CME has been observed in the STAR and PHENIX experiments and cite Ref. [6], however, those observations have not confirmed the CME. There are other interpretations of the observations that are not related to the CME at all. The authors need to state this in the paper and do proper citations. The paper ignores the effects of thermal fluctuations, but thermal fluctuations are not negligible at the temperatures generated in heavy ion collisions, so the authors should give some plausible arguments for the reader to believe their result are valid in the conditions they are proposing. Along the same line, they use the results of Ref. [9] that found the same tendency of increase chiral condensate and decrease of pion mass when T grows, but they mention nothing about other results in the literature that found exactly the opposite trend. The effects of electromagnetic fields are ignored throughout the paper although they are very important for the CME. Will the chiral anomaly term that generates in the presence of mu_5 and electromagnetic fields affect the results of the paper? The authors need to explain whether their finding will be affected by the ongoing literature debate about the validity of the CME. See for instance: JHEP1001, 026 (2010); Phys. Rev. D93 (2016) 105036; and Phys. Rev. D 98 (2018) 074009.

The authors need to address all these issues in their manuscript before the paper can be published.

Round 2

Reviewer 2 Report

Authors properly addressed my previous comments. I recommend publication of the revised manuscript.